# Estimating the impact of trained midwives and upgraded health facilities on institutional delivery rates in Nigeria using a quasi-experimental study design

Karen Ann Grépin [ID],[1] Adanna Chukwuma [ID],[2] Marcus Holmlund [ID],[3] Marcos Vera-Hernandez,[4] Qiao Wang,[5] Pedro Rosa-Dias [ID][6]

For numbered affiliations see end of article.

**Correspondence to**
Dr Pedro Rosa-Dias;
p.rosa-dias@imperial.ac.uk

## ABSTRACT

**Objectives** Studies have shown that demand-side interventions, such as conditional cash transfers and vouchers, can increase the proportion of women giving birth in a health facility in low-income and middle-income countries, but there is limited evidence of the effectiveness of supply-side interventions. We evaluated the impact of the Subsidy Reinvestment and Empowerment Programme Maternal and Child Health Project (SURE-P MCH) on rates of institutional delivery and antenatal care.

**Design, setting and participants** We used a differences-in-differences study design that compared changes in rates of institutional delivery and antenatal care in areas that had received additional support through the SURE-P MCH programme relative to areas that did not. Data on outcomes were obtained from the 2013 Nigerian Demographic and Health Survey.

**Results** We found that the programme significantly increased the proportion of women giving birth in a health facility by approximately 7 percentage points (p=0.069) or approximately 10% relative to the baseline after 9 months of implementation. The programme, however, did not significantly increase the use of antenatal care.

**Conclusion** The findings of this study suggest there could be important improvements in institutional delivery rates through greater investment in supply-side interventions.

## INTRODUCTION

The world has made great advancements in reducing maternal mortality: the estimated number of women dying in childbirth declined from 385 per 100 000 live births in 1990s to 216 in 2015.[1] Progress, however, has not been even and countries in sub-Saharan Africa (SSA) have made the least improvements. Nigeria alone accounted for almost a fifth of the estimated maternal deaths that occurred globally in 2015 and was estimated to have one of the highest maternal mortality ratios in the world at 814 per 100 000 live births.[2]

Over the same time period, the proportion of women giving birth in a health facility had also substantially increased in SSA.[3] However,

## STRENGTHS AND LIMITATIONS OF THIS STUDY

⇒ This project evaluated the impact of a real-world programme that was implemented at scale on the use of maternal health services across Nigeria.

⇒ The programme provided trained midwives and facility upgrades to participating primary health centres.

⇒ Our study combined programmatic data with household survey data to estimate the impact of the programme at the individual level using data from the Nigerian Demographic and Health Survey.

⇒ Due to the reliance on household survey data that were not collected for the purposes of this study, we were limited by sample size and the types of outcomes we could evaluate.

⇒ A lack of precise geographic information on households may have limited our ability to directly attribute effects of the programme.

the lack of commensurate improvements in maternal health outcomes in some countries suggests that the indicators used to monitor maternal health progress may not be fully capturing the elements of the content or of the quality of care received during childbirth that are essential to improve maternal health outcomes.[4] Many of the root causes of maternal mortality have not changed in decades: far too many women are accessing health services 'too little, too late'.[5] And although more women are giving birth in a health facility, and in the presence of a skilled birth attendant (SBA), if those health facilities are not adequately equipped, or if the SBA is not adequately trained and empowered to provide high-quality care, then health outcomes may not improve.

While there is a clear need to improve the quality of maternal healthcare in such settings,[6] there is little rigorous evidence of the impact of such investments on the use of health services. And although studies have

shown that many demand-side interventions (eg, conditional cash transfers and vouchers)[7] can greatly improve the use of health services, there is less rigorous evidence of the effectiveness of supply-side interventions, including those aimed at improving the availability of SBAs and the quality of healthcare facilities.[8 9] Beyond ensuring the presence of a SBA, many factors also limit the ability of SBAs to deliver high quality health services, including lack of training and supervision, excessive workloads, low salaries and poor living conditions, as well as lack of access to well-equipped health facilities. The impact of making improvements in these other dimensions is for the most part not well understood.[10]

Countries in SSA, including Nigeria, rely on a range of cadres of health professionals to provide specialised care to women, infants, and families over the entire continuum of pregnancy from preconception to the early weeks of life and all of these health professionals are classified as SBAs.[11] Midwives and nurses are among the most common types of health professionals that provide skilled assistance across SSA.[12 13] Historically, important reductions in maternal mortality have been linked to the expansion of midwives in high-income countries like Sweden.[14] But less is known about the potential contribution of greater access to midwives in low-income and middle-income countries (LMICs) today. However, a recent modelling study found that the scale-up of midwifery in LMICs could avert a substantial proportion of reproductive-related morbidity and mortality in these countries, although rigorous evidence of scaling up midwives and other related support has not been established.[15]

Building off this gap in the literature, this study aims to evaluate the impact of the Subsidy Reinvestment and Empowerment Programme Maternal and Child Health Project (SURE-P MCH), a large-scale programme introduced in 2012 that deployed trained midwives and was aimed at accelerating progress towards Millennium Development Goals (MDGs) 4 and 5, on institutional delivery (ID) and antenatal care (ANC) rates across Nigeria. Our goal with this study is to generate rigorous evidence on the potential effectiveness of comparable large-scale programmes to improve coverage of high-quality health services in other high maternal mortality settings.

## METHODS
### Study context
In January 2012, the Federal Government of Nigeria eliminated a longstanding fuel subsidy and announced that funds raised by the removal of this subsidy would be used to improve public services, including health services. Following this announcement, the SURE-P MCH programme was launched as a flagship initiative to improve MCH outcomes across the country. The programme was implemented by the National Primary Healthcare Development Agency, a parastatal organisation under the Federal Ministry of Health.

Following its launch, the programme began the process of upgrading primary health centres (PHCs) and training newly recruited midwives. In order to facilitate the rapid roll-out of the programme, facilities were purposively selected for the project based on need (defined as a persistent lack of midwives) and also conditional on meeting a set of minimum infrastructure and human resource requirements. To be eligible for SURE-P MCH, PHCs had to: offer MCH services; have minimum equipment and basic infrastructure, including potable water supply, power supply and sewage disposal; and operate on a twenty-four-hour basis. These criteria, especially the last condition, explain why our treated facilities and their respective catchment areas had higher rates of ID at baseline than our comparison facilities.

Following their training, the programme deployed 1285 midwives to 473 PHCs in high-priority areas across Nigeria's 36 States and the Federal Capital Territory. Each facility usually received more than one additional midwife and the first batch of facilities began to fully participate in the programme in October 2012. The SURE-P MCH programme also benefited from a wide-reaching mass media campaign, encompassing radio and television adverts, billboards and posters encouraging pregnant women to visit SURE-P MCH PHCs after the deployment of the midwives.

Although initially there had been plans to further scale up the programme after the initial phase, unfortunately the programme had been cut short about a year after implementation primarily due a budget shortfall related to a sharp fall in international oil prices. Although the programme was not officially terminated until 2015, news had begun to circulate following the drop in oil prices that the programme would be terminated in early 2014. Programme officials had reported to the research team high levels of midwife attrition, which may have started in late 2013. Due to these factors, we limit our analysis to the first year of programme implementation (October 2012–2013), however, as discussed in the data section, we only had data for the first 9 months of this implementation period to evaluate outcomes.

### Evaluation strategy
To evaluate the impact of SURE-P MCH, we adopted a differences-in-differences (DID) study design, a method that has been widely used to rigorously evaluate the impact of health and social programmes in other comparable contexts.[16] The approach compares the change in outcomes in treated areas to the change in outcomes in comparison areas, attributing any difference in changes to the programme. We define exposure to the programme at the individual level, more specifically we define women who live in areas where SURE-P MCH was implemented to be exposed, which we operationalize with a binary treatment variable. We also define a post-treatment variable, which takes the value of 1 if a birth occurs after the programme was implemented, or 0 otherwise.

**Table 1** Summary of outcomes, characteristics of mothers and households, by study area, prior to the intervention

| | Total | | Treatment | | Comparison | | Difference | |
|---|---|---|---|---|---|---|---|---|
| | N | % | N | % | N | % | Perc. Pt | P value |
| Outcomes | | | | | | | | |
| Delivery assistance: doctor/nurse/midwife | 19 599 | 0.43 | 891 | 0.70 | 18 708 | 0.42 | 0.305 | 0.000 |
| Delivery in any facility | 19 671 | 0.44 | 891 | 0.72 | 18 780 | 0.43 | 0.317 | 0.000 |
| ≥4 times of antenatal visits during pregnancy | 11 533 | 0.62 | 511 | 0.85 | 11 022 | 0.61 | 0.244 | 0.000 |
| Covariates | | | | | | | | |
| Maternal age | 21 755 | 27.62 | 975 | 28.01 | 20 780 | 27.60 | 0.748 | 0.106 |
| Parity at delivery | 21 755 | 2.84 | 975 | 2.24 | 20 780 | 2.87 | −0.613 | 0.000 |
| Covered by health insurance | 14 074 | 0.02 | 648 | 0.03 | 13 426 | 0.02 | 0.004 | 0.592 |
| Currently working | 14 065 | 0.73 | 645 | 0.80 | 13 420 | 0.73 | 0.094 | 0.000 |
| Husband/partner has at least primary school education | 13 594 | 0.71 | 613 | 0.90 | 12 981 | 0.70 | 0.206 | 0.000 |
| Respondent has at least primary school education | 14 136 | 0.63 | 648 | 0.86 | 13 488 | 0.62 | 0.266 | 0.000 |
| Currently married or in union | 14 136 | 0.93 | 648 | 0.91 | 13 488 | 0.93 | −0.025 | 0.026 |
| Respondent is Muslim | 14 075 | 0.53 | 646 | 0.35 | 13 429 | 0.54 | −0.264 | 0.003 |
| Respondent reads newspapers | 14 136 | 0.18 | 648 | 0.29 | 13 488 | 0.18 | 0.104 | 0.014 |
| Respondent listens to radio | 14 136 | 0.66 | 648 | 0.75 | 13 488 | 0.65 | 0.078 | 0.042 |
| Respondent watches television | 14 136 | 0.55 | 648 | 0.77 | 13 488 | 0.54 | 0.229 | 0.001 |
| Household has electricity | 12 519 | 0.59 | 608 | 0.81 | 11 911 | 0.58 | 0.268 | 0.000 |
| Belongs to the poorest two quintiles | 12 634 | 0.32 | 613 | 0.09 | 12 021 | 0.33 | −0.241 | 0.000 |
| Residence is urban | 12 634 | 0.47 | 613 | 0.71 | 12 021 | 0.45 | 0.285 | 0.001 |
| Residence in North East | 12 634 | 0.15 | 613 | 0.05 | 12 021 | 0.15 | −0.097 | 0.000 |
| Residence in North West | 12 634 | 0.26 | 613 | 0.05 | 12 021 | 0.27 | −0.260 | 0.000 |
| Residence in South East | 12 634 | 0.11 | 613 | 0.22 | 12 021 | 0.10 | 0.190 | 0.107 |
| Residence in South South | 12 634 | 0.15 | 613 | 0.15 | 12 021 | 0.15 | 0.035 | 0.631 |
| Residence in South West | 12 634 | 0.17 | 613 | 0.36 | 12 021 | 0.16 | 0.171 | 0.078 |

1. Buffer=100 m and catchment area=2500 m.
2. For birth record level variables, baseline is defined as births prior to October 2012 when the SURE-P MCH programme started.
SURE-P MCH, Subsidy Reinvestment and Empowerment Programme Maternal and Child Health.

For the purposes of this analysis, we consider 1 October 2012 as the start date of the programme, based on information obtained from national programme officials. The effect of the programme is then estimated as the interaction of the treatment variable and the post-treatment variable, conditional on a set of covariates. Our regression model is thus the one below, where X is a vector of covariates listed in table 1:

$$Y_{ict} = \beta_1 SUREp_c + \beta_2 Post_t + \beta_3 SUREp_c * Post_t + \Sigma X_{it} + \varepsilon_{it}$$

In this equation, $Y$ are the outcome variables of interest (IDs or ANC), $SUREp$ is a binary variable that takes the value 1 if the mother ($i$) lives in a household that is located in one of the SURE-P MCH treatment areas, and $Post$ is a time ($t$) dummy variable that takes the value of 1 if the birth occurred after October 2012. X is a set of control variables, which are fully described in our result tables. We also account for sample clustering effects ($c$) in our estimates of CIs and p values.

The DID approach relies on a common trend assumption (ie, parallel trends in the outcome indicators across treatment and comparison groups, in the absence of treatment). However, this assumption can never be formally tested in the presence of an intervention, thus it is standard to test this assumption in the pretreatment period. In the Results section, we provide estimates of our test of the common trend assumption and show that the parallel trends hypothesis cannot be rejected for the entire pretreatment period.

### Data sources

Our data source for the outcome and control variables, which were measured at both the individual and household levels, was the 2013 Nigerian Demographic and Health Survey (NDHS)[17] conducted between February and June 2013. Furthermore, the data on the location of health facilities, obtained from the National MDGs

Information System (NMIS),[18] provides georeferenced locations of over 34 000 health facilities.

In order to define the treatment and comparison status of health facilities, we ascertained the location of all the 473 SURE-P MCH facilities using programme data. We then determined a 100 m radius around each SURE-P MCH facility and defined all facilities within such radius as treatment facilities, under the assumption that all facilities within the same proximity of a treatment facility were indirectly affected by the programme. This produced a final list of treatment facilities in the NMIS, and the remaining facilities were defined as comparison facilities.

In the NDHS, households were sampled within primary sampling units (also called clusters), and these were geocoded. To determine whether each cluster from the 2013 NDHS was within a treatment or comparison area, we first calculated the distance of each cluster to the nearest treatment facility as well as the nearest comparison facility (in metres, using the geocoordinates of the centre of the cluster). We defined treatment clusters as those located less than 2500 m from the nearest treatment facility and comparison clusters as those located within a 2500 m radius of a comparison facility and outside a 2500 m radius of a treatment facility. To avoid comparing households living in very remote locations with those located close to a treatment facility, our analysis excluded a small number of clusters that were located more 2500 m from the nearest treatment facility and over 7500 m from the nearest comparison facility. The selected distances were derived from a literature review on distance-based access to care measures across SSA that accounted for the mean distance of clusters from facilities within the sample.[19] Nonetheless, in our online supplemental appendix, we performed a robustness analysis by changing these thresholds; ultimately our conclusions do not depend on the precise distance of the chosen thresholds chosen.[20]

The two primary outcomes of the study were the rate of ID, defined as the proportion of deliveries, as reported by women, that took place in either a government hospital, health centre, health post or other public sector medical facility, a private hospital, clinic or other medical sector facility; and the percentage of all pregnancies resulting in a live birth for which the mother reported receiving at least four ANC visits (ANC4).

Given that our treatment areas differ from the comparison areas due to the selection criteria of the facilities, a rich set of covariates was also drawn from the DHS dataset including data collected at the individual, household, cluster and regional levels. The covariates were selected based on whether they could have influenced the facility selection criteria and thus represent a potential confounder with the primary outcome variables of interest. A summary of covariates is given in table 1.

### Study registration and changes to protocol

This study was originally registered as an observational study.[21] At the time of registration, it was hoped that we could collect a purposely designed survey in the comparison areas to collect the outcome variables. However, in the end, this was not possible due to a lack of funding. As such, we used the 2013 NDHS as our data source for this evaluation. Among the secondary outcomes that we had previously registered, we could not analyse post-partum depression or pregnancy and obstetric-related healthcare practices because they could not be obtained from the 2013 NDHS.

### Patient and public involvement statement

Patients or the public were not involved in the design, conduct, reporting or dissemination plans of our research.

## RESULTS

Table 1 describes the characteristics of mothers and households in both the treatment and comparisons areas before the start of the intervention. ID rates were much higher in the treatment areas than in comparison areas (72% vs 43%), as was the percentage of births in which the mother had at least ANC4 visits (85% vs 61%). Additional characteristics (such as education and household income) confirm that the treatment areas were significantly better off than the comparison areas. However, it should be noted that the DID method that we employed controls for differences in time-invariant unobservable variables that might affect the outcome of interest if the common trend assumption holds.

Although it is not possible to statistically prove that trends in IDs were parallel prior to the start of the programme, we used hypothesis testing to ascertain whether our data are consistent with the hypothesis of parallel trends in the pre-policy period. Specifically, we performed a Wald test of the hypotheses of common trend assumption and these were never rejected during the pre-policy period years (2008 to 2012) for IDs (p=0.30) and attendance of at least ANC4 visits during delivery (p=0.24). This is also consistent with the assumption, which is true to the best of our knowledge, that there were no other large-scale policies or programmes in place in Nigeria that would have affected the SURE-P MCH facilities differently than those in the comparison areas, which would be difficult given the very specific criteria that were used to select the treatment facilities. We previously discussed these criteria in our Study Context section.

Table 2 shows the estimated programme effects on ID and attendance of at least ANC4 visits 9 months after implementation of the programme. For households within the 2500 m catchment area of a treatment facility, the results in panel 2 show that the programme increased ID rates by 6.7 percentage points and is statistically significant (p=0.069) at the 10% level. And without adjusting for covariates (panel 1), the effect is slightly larger (7.2 percentage points), but the CI is wider. Hence, the covariates improve the precision of the estimates. As for the outcome of ANC4, although the point estimates are positive, they are not statistically significant.

**Table 2** OLS regressions—effect of the intervention on institutional deliveries and use of antenatal care (ANC)

| | DID Coef | 95% CI | P value | N |
|---|---|---|---|---|
| **Panel 1: unadjusted (no controls)** | | | | |
| Institutional delivery | | | | |
| 2000 m | 0.075 | −0.042 to 0.193 | 0.209 | 19 475 |
| 2500 m | 0.072 | −0.032 to 0.175 | 0.175 | 22 343 |
| 3000 m | 0.050 | −0.047 to 0.146 | 0.316 | 24 524 |
| At least 4 ANC visits | | | | |
| 2000 m | 0.048 | −0.034 to 0.131 | 0.254 | 12 279 |
| 2500 m | 0.059 | −0.012 to 0.130 | 0.103 | 14 095 |
| 3000 m | 0.032 | −0.044 to 0.109 | 0.406 | 15 473 |
| **Panel 2: adjusted (with controls)** | | | | |
| Institutional delivery | | | | |
| 2000 m | 0.074 | −0.000 to 0.148 | 0.051 | 18 413 |
| 2500 m | 0.067 | −0.005 to 0.138 | 0.069 | 21 130 |
| 3000 m | 0.050 | −0.008 to 0.107 | 0.090 | 23 240 |
| At least 4 ANC visits | | | | |
| 2000 m | 0.029 | −0.037 to 0.095 | 0.393 | 11 488 |
| 2500 m | 0.030 | −0.028 to 0.087 | 0.311 | 13 200 |
| 3000 m | 0.020 | −0.029 to 0.069 | 0.420 | 14 526 |

1. Control variables include both mother's characteristics (maternal age, square of maternal age, birth order, mother's health insurance coverage, current working status, mother and husband/partner's education level, mother's marriage status, religion and exposure to media) and household characteristics (access to electricity and asset quintiles).
2. All SEs are robust and clustered at DHS cluster level.
DHS, Demographic and Health Survey; DID, differences-in-differences; OLS, Ordinary least squares.

The main estimates were obtained assuming a facility catchment area of 2500 m (as suggested in Okwaraji and Edmond, 2012)[22] and using a radius of 100 m to match SURE-P MCH facilities between the NMIS database and the purposely collected database of SURE-P MCH facilities. For robustness, in table 2, we also provide results for catchment areas of 2000 m and 3000 m. The results are very similar to the main estimates of 2500 m, but slightly larger (7.4 percentage points vs 6.7 percentage points) and with a smaller p value (0.051 vs 0.069) for the catchment area of 2000 m compared with 2500 m. Furthermore, assuming a catchment area of 3000 m compared with that of 2500 m, the estimates are marginally smaller (5.0 percentage points vs 6.7 percentage points) and with a larger p value (0.09 vs 0.069). Also, for robustness, we estimated the model using a facility-matching radius of 200 m instead of 100 m and found nearly identical results.

As a robustness check, we have re-estimated our model for the period October 2012 to October 2013 using all the births recorded in the 2013 NDHS as well additional births from 2013 and 2014 that were captured in the 2018 NDHS. However, only a very small number of additional births (2.8% of the births in the 2018 NDHS) were recorded in 2013. When we again estimate the impact of the programme in the first year, the results are similar to those seen table 2 (and presented in table 1 of the online supplemental appendix). In addition, when we estimate the effects of the programme using a in the second year of implementation, that is, for the October 2013–October 2014, we found no statistically significant effects of the programme on ID, which is consistent with our observation that midwife attrition as well as financial and logistical problems begun to plague the programme after the first year. We therefore prefer the estimates of the first year of implementation using the 2013 Nigerian DHS dataset only.

## DISCUSSION

Improving the quality of maternal health services is an important global health priority for many countries, including Nigeria. In 2012, the Nigerian government launched an ambitious programme that dispatched trained midwives to eligible health facilities across the country. This study found that the increased availability of the midwives led to substantial increases in the proportion of women giving birth in a health facility leading to an increase in ID rates by 7.2 percentage points. This represents approximately a 10% proportional increase in women gaining access to health services, a substantial increase obtained after 9 months of implementation. However, the increased availability of midwives did not cause an increase in the use of ANC.

To contextualise our findings, we compared our findings to those observed in other studies of other programmes aimed at increasing ID rates in similar contexts. Our findings are smaller in magnitude than those found in evaluations of conditional cash transfer programmes in Nigeria.[7 23] However, a recent systematic review of the impact of demand-side programmes on ID rates in low income settings found that financial incentive programmes could increase ID rates on average by 5.3 percentage points, with conditional cash transfer programmes having on average larger effects.[9] Therefore, our from Nigeria findings are comparable to those observed in many demand-side programmes.

While there is limited evidence of the impact of supply-side interventions, a notable exception is a recent study by Croke *et al*, which investigated the impact of national health facility construction programme on delivery rates in Ethiopia using similar data and study design. The authors find similar effect sizes: the construction of a new health facility led lead to a 7.2 percentage point increase in ID rates among treated facilities and the effects were observed almost immediately after the facilities had been constructed.[24] Proportionally our results are smaller, due to higher baseline health service utilisation rates; however, taken together, our study and these findings suggest that supply-side interventions, when properly implemented, can also translate into meaningful gains in ID rates. More research is needed on the complementarity between demand and supply-side policies in this context, as well as on the role of the quality of the services provided.

However, while evidence from this study indicates that supply-side interventions that increase the availability of midwives and upgrade health facilities can have substantial effects in the short-run, it also highlights that it can be challenging to maintain large-scale national-level programmes in many international contexts. More evidence is needed to support the development of programmes aimed at the supply-side and efforts to sustain these programmes over the long-run.

Our study has a few limitations that must be taken into consideration when interpretating our results. First, the DHS data were collected in clusters which were georeferenced but this locational data were displaced to protect the identity of respondents.[25] This means that we may have incorrectly classified some treatment clusters as comparison clusters or vice-versa. However, with this type of measurement error, we would have been more likely to have misclassified treatment clusters as comparison clusters, which should have biased downwards our estimates of the impact of the SURE-P MCH programme. Second, the NMIS database was our only source of geographical information on non-SURE-P MCH facilities and these data were collected in 2012, a year before our outcomes, which could have led to some discrepancies in location as a result. Third, due to our reliance on the NDHS for our key outcomes, our unit of analysis was a birth, not a pregnancy. In other words, we have data on all births, but not necessarily on all pregnances. It is possible that

the differential outcomes between the treatment and comparison groups were also affected by the differential pregnancy termination rates. However, we did not find a significant difference in the proportion of women who had terminated a pregnancy between treatment and comparison areas after the programme started (not shown, but data are available on request). Fifth, we cannot rule out the possibility that women in DHS clusters classified as being in the comparison group responded to the SURE-P MCH programme by seeking care in treatment facilities. However, the procedures used to assign treatment or comparison group status to each DHS cluster means that all comparison clusters were at least 2500 m from treatment facilities, and, based on a literature review on distance-based access to care measures across SSA, we expect access of the comparison group to treatment facilities to be limited.[19] Also, our sensitivity analysis indicates that our results are robust to varying the threshold. Finally, the extent to which the results found here can be replicated within the broader Nigerian primary healthcare system is uncertain. Although SURE-P MCH was implemented in all Nigerian states and the Federal Capital Territory, the facilities selected for SURE-P MCH were, on average, better off in terms of our main outcomes. This was because treatment facilities were partly selected based on the availability of human resources and equipment.

## Conclusion

Following the MDGs, additional resources were channelled towards, and greater focus was placed, on improving maternal health outcomes globally. While the MDGs for MCH were not met in Nigeria, the results of this study demonstrate that supply-side improvements hold promise for increased rates of ID in Nigeria and, likely, other LMICs. Supply-side interventions, which thus far have been poorly studied, represent an under investigated part of the solution to maternal mortality. Therefore, additional research is needed to understand the impact of other supply-side improvements, including the complementary role they can play alongside demand-side incentives, on health outcomes in Nigeria and in other international contexts.

**Author affiliations**
[1]School of Public Health, University of Hong Kong Li Ka Shing Faculty of Medicine, Hong Kong, China
[2]Health, Nutrition, and Population Global Practice, World Bank Group, Washington, DC, USA
[3]Development Impact Evaluation (DIME), World Bank, Washington, District of Columbia, USA
[4]Department of Economics, University College London, London, UK
[5]Water Global Practice, World Bank Group, Washington, District of Columbia, USA
[6]Department of Economics and Public Policy and CHEPI, Imperial College Business School, Imperial College London, London, UK

**Contributors** KAG was involved in the analysis and writing of the original and final manuscripts. AC was involved in the conceptualisation of this project, developing the methodology, administering this project, obtaining funding, reviewing and editing the manuscript. MH was involved in the conceptualisation of this project, developing the methodology, administering this project, obtaining funding,

and managing project resources. MV-H was involved in the conceptualisation of this project, developing the methodology, obtaining funding for the project, administering the project and managing project resources. QW was involved in developing the methodology for the project, curating the data collected for the project, administering the project and providing resources for the project. PR-D was involved in the conceptualisation of the research project, conducting the analysis for the paper, obtaining funding for the project, developing the research project methodology, administering the project, writing the original draft of the paper, reviewing and editing the final version of the manuscript. He is also the guarantor of this study. The funding agency had no role in the drafting of this manuscript.

**Funding**  The World Bank adminsitered Strategic Impact Evaluation Fund (SIEF)

**Competing interests**  Between 2012 and 2014, MV-H and PR-D declare receiving short-term consultancy fees from the World Bank to support this evaluation project. KAG, MH, QW and AC declare no competing interests.

**Patient and public involvement**  Patients and/or the public were not involved in the design, or conduct, or reporting, or dissemination plans of this research.

**Patient consent for publication**  Not applicable.

**Ethics approval**  This study involves human participants and was approved by The University College London Research Ethics Committee (IRB approval number 1827/004, 2013/02/13) and the National Health Research Ethics Committee of Nigeria (approval number NHREC 01012007, 2013/02/02).

**Provenance and peer review**  Not commissioned; externally peer reviewed.

**Data availability statement**  Data are available in a public, open access repository. Data are available on reasonable request. Outcome data are publicly available. All other data are available on request from the authors.

**ORCID iDs**
Karen Ann Grépin http://orcid.org/0000-0003-4368-0045
Adanna Chukwuma http://orcid.org/0000-0001-7873-7633
Marcus Holmlund http://orcid.org/0000-0001-8982-4951
Pedro Rosa-Dias http://orcid.org/0000-0002-5438-4826

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
