## [Reviewer comments · BMJ Open]

ARTICLE DETAILS

TITLE (PROVISIONAL)	Estimating the impact of trained midwives and upgraded health facilities on institutional delivery rates in Nigeria using a quasi-experimental study design
AUTHORS	Grépin, Karen; Chukwuma, Adanna; Vera-Hernandez, Marcos; Wang, Qiao; Rosa-Dias, Pedro

VERSION 1 – REVIEW

REVIEWER	Naveen Sunder Bentley University, Economics
REVIEW RETURNED	14-Jul-2021

GENERAL COMMENTS	The impact of trained midwives and upgraded health facilities on institutional delivery rates in Nigeria Detailed Reviews 1. The authors examine the impact of the policy in the immediate aftermath of its implementation – the policy was implemented between September 2012 and May 2013, whereas the outcome data is based on a survey conducted in the months between February and May 2013. This implies that my prior (before looking at the results) would be that one would find null effects. This reasoning is largely driven by the following intuition – large scale national policies implemented in resource-poor contexts (like Nigeria) typically experience delays and teething issues. Therefore, it might take a while before the policy is fully effective and people can take full benefit of it. a. The authors show that the rate of institutional birth went up by 7 p.p. (around 16 percent of the mean) in response to this policy. Such a large effect seems implausible in such a short period of time. Additionally, the documented effects would be plausible only if people living around the facilities were able to gauge the improvement in quality and change their behavior in a rapid manner. This seems unlikely, especially in the rural Nigerian context. b. There is DHS data from Nigeria available for the year 2018. Is there any particular reason the authors do not use this? I ask because using that dataset might help increase the number of years of the outcome that the authors can examine. Basically, they can use birth histories provided by mothers to look at the effects of the policy on outcomes in 2014, 2015 etc. 2. The authors have some excellent information on location of health facilities across the country that they combine with the DHS data. I am not totally sure that they are doing the best they can. Instead of directly critiquing their process, I provide an alternative procedure that the authors can use to merge these datasets (which I
---

	think maybe better). I would like the authors to consider this, and in their response comment on why this may or may not be the best thing to implement:  a. Download a shape file that provides boundaries of administrative divisions at the sub-district level (source = https://www.diva-gis.org/gdata) b. Take all the facility locations, and map it to a sub-district in step a. Now you can calculate the number of facilities in each sub-district in the country. c. Repeat step b, but now using the DHS GPS data. That is, merge DHS GPS data with the file in step a. 3. I suggest point 2 above, because I am not totally convinced by the catchment area that you assume (2000-3000 mts).  a. The results are fairly sensitive to the range you choose. In panel 2 of table 2, the p-value falls as the distance is increased. b. Catchment area depends on terrain. It might be hard to travel large distances in places with mountains, but it may be fairly easy to go even 15-20 kms in urban areas. 4. Instead of using the treatment status of the closest PHC in the data, I suggest that the author(s) think of the following alternative: Based on the dataset created in (2) above, one can assign the treatment status of an individual in the DHS data as the following: # treatment facilities in the district (or sub-district) divided by total # of facilities in the district (or sub-district). This will assign each cluster in DHS a number between 0 and 1, where a higher number is a more *intensive* treatment. In my opinion, this strategy deals with two important issues with the analysis:  a. Individuals could travel larger distances (> 3000 mts) to other facilities. This strategy accounts for that as we look at the proportion of treatment facilities in a larger catchment area. b. Mitigates some of the issues that arise with the scrambled GPS location in the DHS data. The authors should use this new *treatment* variable in their empirical strategy and see if they get similar results. If they feel that this is not justified, then I would be happy to hear why this strategy is not suitable in this context. 5. The discussion of parallel trends is unsatisfactory at best.  a. The scale of the graph in Figure 1 is deceptive and gives the illusion of parallel trends. Can you present graphs where the y-axis varies between 0.4 and 1 ? 6. Can you present results with district fixed effects? This would make the readers *believe* the results more. 7. Can the authors please write their empirical strategy in the form of an equation? 8. Table 1 – the N varies a lot across variables. Is this a typo, or are there a lot of missing values. 9. Lines 19-26, Page 13 – the reasoning provided here is unclear.
--	---

REVIEWER	Julius Emmrich Charite Universitätsmedizin Berlin
REVIEW RETURNED	15-Dec-2021

GENERAL COMMENTS	This is an interesting and well-written work describing the impact of trained midwives and upgraded health facilities on health outcomes in Nigeria. Using a differences-in-differences design, the authors find that the intervention increases the rate of facility-based deliveries but not utilization of antenatal care.
---

	Major:  - My major concern is the large difference in outcome levels between intervention and comparison areas before the start of the intervention. In general, a DiD is more plausible if outcome levels - not just trends - are similar before the start of an intervention. The authors fail to address in sufficient detail why the original levels of the intervention and comparison areas differ and why one shouldn't think these same mechanisms would not impact future trends. Moreover, the authors fail to discuss why they think that under these circumstances it is reasonable to assume that the parallel trends assumption is justified. The authors imply treatment areas being "better off than comparison areas" for indicators like education and income. The authors might want to consider controlling for these factors in their analysis for example by using a matched sample, as this would greatly strengthen the relevance of their findings. Minor:  - It would be unusual for a health systems intervention like the one described here to be implemented at all facilities at once; one would rather expect a sequential implementation. Taking into account the very short overall program duration of only 9 months, the authors might want to describe the exposure status of facilities during the program in more detail (i.e. what was the average, min/max duration of implementation)? - The authors state that all facilities within a 100m radius around intervention facilities were categorized as treatment facilities, the remaining facilities were defined as comparison facilities. Could the authors be more specific on whether "remaining facilities" refers to all 34,000 health facilities included in Nigeria's MIS, which were not exposed to the intervention? If so, Fig 1 certainly does not show all NHMIS comparison facilities. - There are numerous typos and other editing mistakes throughout the manuscript. The authors might want to consider giving it a careful read through before re-submission. Some examples: Abstract: "... facility across in low and middle-income countries..." Intro: "protentional contribution" Methods: "We defined treatment clusters those located..." Use of abbreviations: Some lack to be defined at first mention Decimal separators used inconsistently
--	---

VERSION 1 – AUTHOR RESPONSE

Reviewer: 1

The authors examine the impact of the policy in the immediate aftermath of its implementation – the policy was implemented between September 2012 and May 2013, whereas the outcome data is based on a survey conducted in the months between February and May 2013. This implies that my prior (before looking at the results) would be that one would find null effects. This reasoning is largely driven by the following intuition – large scale national policies implemented in resource-poor contexts (like Nigeria) typically experience delays and teething issues. Therefore, it might take a while before the policy is fully effective and people can take full benefit of it.

- ***What can we say about the roll out operationally? Do we know when people were deployed?***
- ***Could we restrict our data to a time period towards the enter of the intervention period to see if we get stronger effects compare to those in the earlier time period?***

We thank the Reviewer for raising this issue, and agree that this was not sufficiently discussed in the original version of the manuscript. Indeed, interventions in resource-poor countries are often rolled-out sluggishly and are not advertised sufficiently to reach their potential beneficiaries. The combined effect of these factors often delays policy effects, as rightly pointed-out by the Referee.

However, the context in which this program was designed and rolled-out was, with regards to both these factors, an exception. In January 2012, the Federal Government of Nigeria announced the removal of an important fuel subsidy that had been in place for many years, causing the retail prices of petrol to instantly double. This prompted a wave protests among the general public throughout the country. The government countered that the primary purpose of removing this subsidy was to invest these funds into areas such as the healthcare system. Thus, the SURE-P – MCH programme was announced soon after the removal of the subsidy as a flagship initiative and was heavily promoted to appease the protesters and other critics. Given this motivation, this translated into three key important aspects of the programme's design.

First, the program was initially well-funded, as this was key for demonstrating that the funds obtained through the removal of the subsidy were being reinvested. Second, SURE-P MCH benefited from a mass media campaign, encompassing radio and television adverts, billboards, and posters encouraging pregnant women to visit SURE-P MCH primary health centres. This campaign also had the objective of making clear that the improvement of SURE-P MCH PHCs was due to government policy, thereby justifying the highly unpopular fuel subsidy removal. The government was committed to this campaign, which was well-funded and wide-reaching. Third, the criteria used to select the first 500 PHCs to receive the intervention were also informed by the government's preference for a quick roll-out, that made the benefits of reinvesting the proceeds of fuel subsidy removal salient to the communities. Thus, to qualify to receive SURE-P MCH, healthcare facilities had to already offer maternal and child health service; have minimum equipment and basic infrastructure including potable water supply, power supply, and sewage disposal; operate in on a twenty-four-hour basis. This also helps to explain why our treated facilities were better at baseline than our comparison facilities.

In addition, the upgrading of health facilities and training of midwives began soon after the announcement of the programme and the first batch of facilities began to fully participate in the programme in October 2012 when the midwives had been deployed. Based on the above factors, as well as the fact that many of the members of the research team had been present during the early phase of the programme, we are confident that the programme was widely known to the programme and the facilities prepared to provide additional services when it launched. In the revised version we have added a few additional paragraphs at the start of the Study Context section of the manuscript, to provide a better description of the timing and promotion of the programme. We note that the language we use in the text of the article was somewhat less politically descriptive than the language we share with the reviewers above.

We also appreciate the reviewer's suggestion to restrict the analysis to an early vs late time period, but as the reviewer has also pointed out, we are already dealing with a relatively short time period, thus limiting our power to be able to say something meaningful about births that happened during the

early implementation phase vs those that happened later. Plus given the pace of implementation described above it also not clear that there is sufficient variation in the time of implementation (and if there were a few outliers, those would probably be brought about by factors that might simultaneously affect policy impact). We discuss the addition of additional data from the 2018 NDHS below.

The authors show that the rate of institutional birth went up by 7 p.p. (around 16 percent of the mean) in response to this policy. Such a large effect seems implausible in such a short period of time. Additionally, the documented effects would be plausible only if people living around the facilities were able to gauge the improvement in quality and change their behavior in a rapid manner. This seems unlikely, especially in the rural Nigerian context. Could we compare this to results from other studies to show that they are in a plausible range to those that have been observed previously?

Again, we thank the Reviewer for this comment and refer again to our discussion to his previous question above. However, we think this question about the magnitude of the response is another very good point. We have thus dug into the literature to develop a response to this question. However, given that there are limited studies that have investigated the effects of supply side interventions on the use of health services in a rigorous manner, we are somewhat limited in terms of the studies we can draw from to inform an answer to this question.

One notable exception is a recent study by Croke et al, which also used DHS data and a difference-in-difference study design, investigated the impact of health facility construction on delivery rates in Ethiopia. Facility delivery rates are very low by international standards in Ethiopia and the construction of a health facility is a more visible intervention than scaling up the number of health workers present at a health facility. Despite those caveats, the authors find very large effects: the construction of a new health facility can lead to a 7.2 percentage point increase in facility delivery rates and an increase in 0.38 antenatal visits within 5 kms of the new health facility. When they restrict their analysis to district hospitals (a higher-level health facility) the effects were even larger: it led to an 18.2 percentage point increase in facility delivery rates. The timeframe over which they observe effects is up to four years post facility construction, however, based on their event study analysis, in the case of facility delivery, the biggest jump occurs “immediately” at time zero and then does not substantial increase afterwards. Interestingly, while this study also identified very large effects, it did not find any effect on neonatal or child mortality.

Full citation: Croke, K., Mengistu, A. T., O’Connell, S. D., & Tafere, K. (2020). The impact of a health facility construction campaign on health service utilisation and outcomes: analysis of spatially linked survey and facility location data in Ethiopia. *BMJ Global Health*, 5(8), e002430. <https://doi.org/10.1136/bmjgh-2020-002430>

In addition, there have been numerous studies that have aimed to investigate the impact of demand side incentives, mainly financial incentives (vouchers, conditional cash transfers, etc.) and financial incentives to providers on the use of health services. A recent and notable systematic review by Neelsen et al has reviewed this evidence for us, which we summarize here. It is worth noting, as the authors of this systematic review correctly point out, that it is challenging to compare effect sizes across the diversity of studies included in this review, as they vary in terms of the baseline health service utilization rates of the countries investigated, the types of schemes investigated, and the follow-up time periods of the studies.

With regards to facility delivery rates, our main outcome variable, this systematic review found on average that the financial incentive schemes investigated could increase rates of institutional deliveries on average by 5.3 percentage points, although they not that there was moderately high level of variation across the schemes. PBF schemes had the smallest effect sizes, on average only 4.4 percentage points, followed by voucher schemes, which had a mean effect size of 6.4 percentage points, and then conditional cash transfer schemes at 7.3 percentages points. While it is hard to summarize the length of the studies investigated, many did look at relatively short time frames (e.g. during the length of a pregnancy) and thus it is plausible that we might find similar effects in our sample. They find comparable estimates when the outcome variable is restricted to delivery in a facility and in the presence of a skilled provider (Figure A7.5).

With regards to antenatal care, the effects tend to be relatively small. They conclude that the studies investigated increased the probability that women received four or more antenatal care visits was just 1.4 percentage points, but it was significant. Within this range, results-based financing schemes had almost no effect, vouchers had a moderate effect, and conditional cash transfers were the more effective.

Full citation: Neelsen, S., Walque, D. de, Friedman, J. & Wagstaff, A. Financial Incentives to Increase Utilization of Reproductive, Maternal, and Child Health Services in Low- and Middle-Income Countries: A Systematic Review and Meta-Analysis. *Policy Res Work Pap* (2021) doi:10.1596/1813-9450-9793.

Given the ranges observed in earlier related studies, it seems plausible that an intervention such as SURE-MCH, which greatly increased the capacity of health facilities to supply skilled birth attendants and that was accompanied by a mass media campaign, could have led to large and immediate increases in facility delivery rates. Or at least, this literature review would suggest that our results are not outliers or inconsistent with previous studies. We have reworked the discussion section on to better discuss our findings relative to those observed in other studies. We thank the reviewer for this suggestion to strengthen the interpretation of our findings.

There is DHS data from Nigeria available for the year 2018. Is there any particular reason the authors do not use this? I ask because using that dataset might help increase the number of years of the outcome that the authors can examine. Basically, they can use birth histories provided by mothers to look at the effects of the policy on outcomes in 2014, 2015 etc.

There are a number of reasons why we had not initially included data from 2018, which we describe below. However, based on our discussion below, it is still our preference to exclude these data from the analysis, although we have followed the Reviewer's suggestion as a robustness exercise.

First, having been present during the implementation of the programme, we had observed that the programme had begun to experience implementation challenges about a year after the programme had been launched. For example, we know that by August 2014, when we formally investigated this

issue, we had found that a significant number of midwives had already left their post, suggesting that implementation of the programme had begun to deteriorate, most certainly throughout most of 2014 and likely part of 2013. It was for this reason, we had initially planned to focus our evaluation on only the first year of implementation.

Second, initially we had planned a follow-up survey to formally test the impact of the intervention, however, once the funding for the programme had been pulled, this was no longer feasible, thus we had to rely upon secondary sources of data, namely the NDHS. As the reviewer points out, the 2013 NDHS only covered 9 months of programme implementation, limiting our analysis, however, it contained a large number of observations over these months. Although the 2018 NDHS also includes births from 2013, the survey did not actually start until late in 2018. As such, there are only a very small number of births (2.8% of births) in the survey from 2013, most in the last few months of 2013. Plus, since births in the 2013 survey were more recent and births from the 2018 survey were distant, there might be some differential recall bias introduced by using these additional births in our sample.

Nonetheless, below we followed the Reviewer’s suggestion of adding the births in 2018 NDHS for the years of 2013 and 2014, as a robustness exercise. The results, presented below for the reviewers’ benefit, corroborate the policy effect in the first year of implementation (our estimates are very close to the ones in the main paper) and also documents the lack of an effect of the programme in the second year of implementation, after the initial implementation impetus faded and attrition and lack of funding affected the program. In the revised version we discuss this in tandem with other robustness checks at the end of our Results section.

Given the fact that adding the data from the more recent survey does not alter our main findings, it is our preference to stick with our initial dataset and results, but we have now added a discussion of this robustness check in the manuscript.

First year of SURE-P MCH : Oct 2012 - Oct 2013 (with controls)

	DID Coef	95% Confidence Interval		P-value
Institutional delivery				
2000m	0.079	0.00	0.15	0.04
2500m	0.068	0.00	0.14	0.06
3000m	0.045	-0.01	0.10	0.14
At least 4 times of ANC visits				
2000m	0.031	-0.03	0.10	0.35

2500m	0.032	-0.02	0.09	0.26
3000m	0.002	-0.06	0.06	0.95

Second (final) year of SURE-P MCH: Oct 2013 - Oct 2014 (with controls)

	DID Coef	95% Confidence Interval		P-value
Institutional delivery				
2000m	-0.015	-0.14	0.02	0.16
2500m	0.019	-0.10	0.07	0.67
3000m	-0.015	-0.09	0.06	0.68
 At least 4 times of ANC visits				
2000m	-0.078	-0.19	0.04	0.18
2500m	0.052	-0.15	0.04	0.29
3000m	-0.049	-0.13	0.04	0.26

Was there any scale-up of the programme after the fact in the comparison regions, if not then I think at the very least we could try this approach to see if we still get some effects in the treatment vs comparison groups in say 2014 and 2015 as well.

As we now better describe in the manuscript, this programme was never scaled-up. In fact, after the initial year of implementation the programme began to deteriorate. Oil prices dropped dramatically in early 2014 causing funding issues, severe midwife attrition weakened the programme, and rumours began to spread that the programme would not be renewed. In fact, 2014 was the last year in which the programme was funded. In addition, immediately after the election of President Buhari, in early 2015, the government announced that SURE-P MCH be cancelled. Some of the SURE-P MCH facilities may have received some funding from international donors, but certainly there were not federal outreach efforts to communities to encourage them use SURE-P MCH facilities. Finally, these interventions from international donors are also likely to have benefitted control PHCs, further confounding the analysis of the original policy impacts from 2015 onwards.

While we cannot say for sure when the implementation of the programme really began to decline, we believe that implementation had already begun to decline in late 2013 or early 2014 and thus our preference is to continue to only consider births in 2013 as treated. Nonetheless, as mentioned in the previous point, we followed the Reviewer's suggestion of adding the births in 2018 NDHS for the years of 2013 and 2014, as a robustness exercise. As mentioned above (and discussed in the revised version) this has confirmed the efficacy of the programme in its first year implementation and corroborated its erosion in 2014.

The authors have some excellent information on location of health facilities across the country that they combine with the DHS data. I am not totally sure that they are doing the best they can. Instead of directly critiquing their process, I provide an alternative procedure that the authors can use to merge these datasets (which I think maybe better). I would like the authors to consider this, and in their response comment on why this may or may not be the best thing to implement:

a. Download a shape file that provides boundaries of administrative divisions at the sub-district level (source = <https://www.diva-gis.org/gdata>)

b. Take all the facility locations, and map it to a sub-district in step a. Now you can calculate the number of facilities in each sub-district in the country. c. Repeat step b, but now using the DHS GPS data. That is, merge DHS GPS data with the file in step a.

3. I suggest point 2 above, because I am not totally convinced by the catchment area that you assume (2000-3000 mts). a. The results are fairly sensitive to the range you choose. In panel 2 of table 2, the p-value falls as the distance is increased. b. Catchment area depends on terrain. It might be hard to travel large distances in places with mountains, but it may be fairly easy to go even 15-20 kms in urban areas.

4. Instead of using the treatment status of the closest PHC in the data, I suggest that the author(s) think of the following alternative: Based on the dataset created in (2) above, one can assign the treatment status of an individual in the DHS data as the following: # treatment facilities in the district (or sub-district) divided by total # of facilities in the district (or sub-district). This will assign each cluster in DHS a number between 0 and 1, where a higher number is a more *intensive* treatment. In my opinion, this strategy deals with two important issues with the analysis:

This is another excellent suggestion from the reviewer. However, we were unable to implement this strategy based on the following issues we encountered:

- There are 775 sub-districts in Nigeria according to the GPS file provided, in Nigeria these units are called LGA or local government areas. As we note in the text, there were only 473 SURE-P facilities in the programme.
- The selection of health facilities first occurred at the geographic level, meaning that there were some LGAs that were first selected for inclusion into the study and others were not.
- Based on the data on our SURE-P facilities, only 119 LGA had at least one treatment facility, however, there was little variation in the number of treatment facilities within each LGA: 90%

of the LGAs had received 4 facilities and only 5% had 3 or fewer, and 5% had 8. Thus, there is almost no variation in number of facilities by geographic region.

As such, the method suggested above by the reviewer would not allow for us to get much variation in exposure to treatment facilities.

The discussion of parallel trends is unsatisfactory at best

We agree with the reviewer: this should be discussed in greater detail. Our difference-in-differences approach requires that in the absence SURE-P MCH, the trends in the levels of ID and ANC be parallel. The Reviewer correctly points-out that graphical analyses provide only suggestive evidence that this is the case. In the revised version of the paper, we discuss the plausibility of this assumption from two additional perspectives. The first is heuristic. Given that SURE-P MCH was implemented in all states of Nigeria, this assumption would fail if another policy or equally wide-reaching phenomena affected asymmetrically treated and control PHCs (and / or their catchment areas). To the best of our knowledge no such policy took place, in Nigeria, during this time period. This would also be implausible given the very particular criteria informing the choice of the first PHCs to receive the intervention (this is discussed above).

The second perspective under which we discuss this issue in the revised version is statistical. Although one cannot that prove that trends are parallel, one can use hypothesis testing to ascertain that our data are consistent with the hypothesis of parallel trends. Although this was not included in the main text of our original manuscript, we performed Wald test of the hypotheses of common trend assumption and these were never rejected during the pre-policy period years (2008 to 2012) for institutional delivery (P-value = 0.30) and attendance of at least four antenatal visits during delivery (P-value = 0.24). In the revised version of the manuscript, this explanation is given at the beginning of our Results section.

The scale of the graph in Figure 1 is deceptive and gives the illusion of parallel trends. Can you present graphs where the y-axis varies between 0.4 and 1 ?

We also thank the Reviewer for this suggestion. In the revised version, the Y-axis of Figure 1 varies between 0.4 and 1, as suggested.

Can you present results with district fixed effects? This would make the readers *believe* the results more.

The number of treated facilities is insufficient to carry-out a meaningful analysis at the district or even state level: the overwhelming majority of these have none or a very small number of SURE-P MCH facilities (each of which has a generally low number treated births). Nonetheless, we have included included fixed effects for geo-political zones in our regression analysis, although this had not been emphasised in the original manuscript.

In Nigeria, geopolitical zones correspond approximately zones of the territory that are relatively homogeneous in terms of ethnic groups. In the context of health services, these are often used

because federal resources for health (and public services more generally) are often shared between these. In the revised manuscript this was made clear in the second paragraph of our Results section.

Can the authors please write their empirical strategy in the form of an equation?

We agree that this makes our empirical strategy clearer to the reader and have included it in the Methods section. In the revised version, this equation is presented on page XZY.

$$Y_{it} = \alpha_{it} + \beta_1 SUREp_{it} + \beta_2 Post_t + \beta_3 SUREp_{it} * Post_t + \Sigma X_{it} + \varepsilon_{it}$$

A description of the equation was also added to the manuscript.

Table 1 – the N varies a lot across variables. Is this a typo, or are there a lot of missing values.

Yes, there are missing values due to item non-response, which is common in the DHS.

Lines 19-26, Page 13 – the reasoning provided here is unclear.

We believe the reviewer is referring to the following line:

“First, the DHS data were collected in clusters which were geo-referenced, but this locational data is displaced to protect the identity of respondents. This means that we may

have incorrectly classified some treatment clusters as comparison clusters, and vice-versa.

However, with this type of measurement error, we would have been more likely to have

misclassified treatment clusters as controls, which should have bias downwards our estimates

of impact as the SURE-P programme, not the other way.”

As there are relatively small number of treatment facilities and thus the majority of area in Nigeria would be classified as a comparison area in our dataset. Thus, if we misclassify a treatment cluster, it is likely it would be incorrectly classified as a comparison cluster. However, the opposite is not true. If we mis-identify the location of a treatment cluster, then it is most likely still counted among the treatment clusters.

With this form of classification bias, more treatment clusters will be mis-classified as comparison clusters, which did not receive the additional midwives and upgrades, and thus this should bias our estimates downwards.

In the revised version, we have replaced our original explanation (which we agree was potentially confusing) by this one, in the Discussion section.

Reviewer: 2

Dr. Julius Emmrich, Charite Universitätsmedizin Berlin

This is an interesting and well-written work describing the impact of trained midwives and upgraded health facilities on health outcomes in Nigeria. Using a differences-in-differences design, the authors find that the intervention increases the rate of facility-based deliveries but not utilization of antenatal care.

We thank the reviewer for this summary of the paper.

My major concern is the large difference in outcome levels between intervention and comparison areas before the start of the intervention. In general, a DiD is more plausible if outcome levels - not just trends - are similar before the start of an intervention. The authors fail to address in sufficient detail why the original levels of the intervention and comparison areas differ and why one shouldn't think these same mechanisms would not impact future trends. Moreover, the authors fail to discuss why they think that under these circumstances it is reasonable to assume that the parallel trends assumption is justified. The authors imply treatment areas being "better off than comparison areas" for indicators like education and income. The authors might want to consider controlling for these factors in their analysis for example by using a matched sample, as this would greatly strengthen the relevance of their findings.

We thank the Reviewer for raising two important points that, we agree, were not satisfactorily explained in the original manuscript.

The first is the reason why treatment areas fare better than control areas in terms of the relevant outcomes. The explanation for this has to do with rationale informing the selection of the first facilities where the programme was implemented, which needs to be contextualised .

In January 2012, the Federal Government of Nigeria announced the removal of an important fuel subsidy that had been in place for decades, causing the retail prices of petrol to instantly double. This prompted a wave protests among the general public throughout the country. The government countered that the primary purpose of removing this subsidy was to invest these funds into areas such as the healthcare system. Thus, *Subsidy Reinvestment and Empowerment Programme Maternal and Child Health* (SURE-P MCH) programme was launched as a flagship initiative and was heavily promoted to appease the protesters and other critics. Given this motivation, this translated into important aspects of the programme's design.

First, the program was initially well-funded and benefited from a mass media campaign, with the goal of making clear that the improvement of SURE-P MCH PHCs was due to government policy, thereby justifying the highly unpopular fuel subsidy removal. Second, and most important to address the Reviewer's point, the criteria used to select the first 500 PHCs to receive the intervention were informed by the government's preference for a quick roll-out, that made the benefits of reinvesting the proceeds of fuel subsidy removal salient to the communities.

To qualify to receive SURE-P MCH, healthcare facilities had to already offer maternal and child health service; have minimum equipment and basic infrastructure including potable water supply, power supply, and sewage disposal; operate in on a twenty-four-hour basis. This automatically caused the treated facilities to be generally better staffed, equipped and to have better infrastructure than the control ones. As these characteristics are positively correlated with relatively less needy communities, SURE-P MCH facilities and their catchment areas are systematically better than the control ones, as pointed-out by the reviewer.

In the revised version this issue is thoroughly discussed in the beginning of our Results section.

The second issue rightly pointed-out by the reviewer is the lack of emphasis given in our original manuscript to the way in which our analysis controls for these large differences in observable characteristics between treatment and control clusters. The Reviewer suggests matching or an equivalent approach for controlling observables.

In fact, although our original manuscript was not entirely clear, our analysis already controls for a rich set of observables in our regression analysis (which is in this context equivalent to matching), namely household characteristics (assets, access to water and electricity, health insurance coverage), mother's characteristics (mother's marriage status, maternal age, square of maternal age, birth order), as well as mother and husband/partner's education level, current working status, religion, and exposure to media. In addition, our regression analysis includes fixed effects for geo-political zones of Nigeria i.e. zones of the country that are relatively homogeneous in terms of ethnic groups and amongst which federal resources for public services (such as health) are usually shared.

In addition to directly controlling for a rich set of covariates, our difference-in-differences approach also controls for differences in time-invariant unobservable variables that might affect the outcomes of interest. In the revised version we have included a discussion of our approach to controlling for observable and unobservable confounders in the Results section.

Minor:

It would be unusual for a health systems intervention like the one described here to be implemented at all facilities at once; one would rather expect a sequential implementation. Taking into account the very short overall program duration of only 9 months, the authors might want to describe the exposure status of facilities during the program in more detail (i.e. what was the average, min/max duration of implementation)?

The Reviewer makes a good point, as the roll-out of this type of programme is often protracted and asynchronous in LMICs. SURE-P MCH is however an exception to this rule. As mentioned in our previous point, this programme was a flagship initiative, heavily promoted by the Federal Government of Nigeria to compensate critics of the removal of a longstanding subsidy fuel subsidy in the country. This required a rapid policy roll-out, that would ideally be salient to the communities.

The purposeful selection of the first batch of facilities to receive the programme, discussed in the previous point, accelerated policy roll-out. In addition, SURE-P MCH benefited from a mass media campaign, encompassing radio and television adverts, billboards, and posters encouraging pregnant women to visit SURE-P primary health centres. This campaign, that also aimed at making the “subsidy reinvestment” salient to the public, also contributed for the programme to reach the communities faster than in other cases.

The upgrading of health facilities and training of midwives began soon after the announcement of the programme and the first batch of facilities began to fully participate in the programme in October 2012. In the revised version we have added a discussion of this Study Context section, to provide a better description of the timing and promotion of the programme. We note that the language we use in the text of the article was somewhat less politically descriptive than the language we share with the reviewers above.

The authors state that all facilities within a 100m radius around intervention facilities were categorized as treatment facilities, the remaining facilities were defined as comparison facilities. Could the authors be more specific on whether "remaining facilities" refers to all 34,000 health facilities included in Nigeria's MIS, which were not exposed to the intervention? If so, Fig 1 certainly does not show all NHMIS comparison facilities.

Upon reflection we agree with the referee that the insufficient granularity in Figure 1 limits the usefulness of this graph, so we have removed it from the revised version of the paper.

There are numerous typos and other editing mistakes throughout the manuscript. The authors might want to consider giving it a careful read through before re-submission. Some examples:

We have also had the manuscript reviewed by a research assistant with a background in journalism and editing, who we believed has found many important typos and has improved the text.

Abstract: “... facility across in low and middle-income countries...”

This text has now been replaced and thus addressed.

Intro: “protentional contribution“

This typo has been fixed.

Methods: “We defined treatment clusters those located...”

This typo has been fixed.

Use of abbreviations: Some lack to be defined at first mention

We have tried to address these issues in the revised text.

Decimal separators used inconsistently

We have now added a decimal for all no-date numbers greater than 3 digits.

VERSION 2 – REVIEW

REVIEWER	Naveen Sunder Bentley University, Economics
REVIEW RETURNED	21-Mar-2022
GENERAL COMMENTS	Thank you for addressing the comments in a satisfactory manner - the manuscript is now much improved, and would be a good addition to the literature. Looking forward to seeing it in print!